# Genome-wide association studies and fine-mapping identify genomic loci for n-3 and n-6 polyunsaturated fatty acids in Hispanic American and African American cohorts

Chaojie Yang [1,2], Jenna Veenstra[3], Traci M. Bartz[4,5], Matthew C. Pahl[6,7], Brian Hallmark [8], Yii-Der Ida Chen [9], Jason Westra[10], Lyn M. Steffen[11], Christopher D. Brown[12,13], David Siscovick[14], Michael Y. Tsai[15], Alexis C. Wood[16], Stephen S. Rich [1], Caren E. Smith[17], Timothy D. O'Connor [18], Dariush Mozaffarian [19], Struan F. A. Grant [6,7,12,20,21], Floyd H. Chilton[22], Nathan L. Tintle [10,23], Rozenn N. Lemaitre [5] & Ani Manichaikul [1✉]

Omega-3 (n-3) and omega-6 (n-6) polyunsaturated fatty acids (PUFAs) play critical roles in human health. Prior genome-wide association studies (GWAS) of n-3 and n-6 PUFAs in European Americans from the CHARGE Consortium have documented strong genetic signals in/near the *FADS* locus on chromosome 11. We performed a GWAS of four n-3 and four n-6 PUFAs in Hispanic American ($n = 1454$) and African American ($n = 2278$) participants from three CHARGE cohorts. Applying a genome-wide significance threshold of $P < 5 \times 10^{-8}$, we confirmed association of the *FADS* signal and found evidence of two additional signals (in *DAGLA* and *BEST1*) within 200 kb of the originally reported *FADS* signal. Outside of the *FADS* region, we identified novel signals for arachidonic acid (AA) in Hispanic Americans located in/near genes including *TMX2*, *SLC29A2*, *ANKRD13D* and *POLD4*, and spanning a > 9 Mb region on chromosome 11 (57.5 Mb ~ 67.1 Mb). Among these novel signals, we found associations unique to Hispanic Americans, including rs28364240, a *POLD4* missense variant for AA that is common in CHARGE Hispanic Americans but absent in other race/ancestry groups. Our study sheds light on the genetics of PUFAs and the value of investigating complex trait genetics across diverse ancestry populations.

A full list of author affiliations appears at the end of the paper.

Omega-3 (n-3) and omega-6 (n-6) polyunsaturated fatty acids (PUFAs) are critical structural components of cell membranes, which can influence cellular activities by promoting the fluidity, flexibility, and the permeability of a membrane[1–3]. In addition, PUFAs affect a variety of other biological processes and molecular pathways, including modulating membrane channels and proteins, regulating gene expression through nuclear receptors and transcription factors, and conversion of the PUFAs themselves into bioactive metabolites[4]. Levels of circulating PUFAs and long chain (≥20 carbons) PUFAs (LC-PUFAs) are associated with reduced risk of cardiovascular disease[5,6], type 2 diabetes mellitus[7], cognitive decline[8], Alzheimer's disease[9], metabolic syndrome[10] and breast cancer[11], as well as all-cause mortality[12].

PUFAs and LC-PUFAs are characterized by the position of the first double bond from the methyl terminal (omega; ω; or n−FAs) and fall into two primary families, n-3 and n-6. The most abundant n-3 PUFAs are alpha-linolenic acid (ALA), eicosapentaenoic acid (EPA), docosapentaenoic acid (DPA) and docosahexaenoic acid (DHA), while the primary n-6 PUFAs are linoleic acid (LA), gamma-linolenic acid (GLA), dihomo-γ-linolenic acid (DGLA) and arachidonic acid (AA). ALA and LA are essential n-3 and n-6 PUFAs consumed from the diet and these then can be converted to more unsaturated LC-PUFAs through a set of desaturation and elongation enzymatic steps. For example, DGLA and AA can be synthesized from LA, while EPA, DPA, and DHA can be produced from ALA. The precursors LA and ALA are essential fatty acids that must be provided by the diet. Due to the lower abundance of ALA in Western diets and the inefficiency of conversion of ALA to longer chain n-3 LC-PUFAs such as EPA and DHA, dietary intake of these via fatty fish or marine oil supplementation is often recommended[13,14].

Previous studies have shown that African ancestry populations have higher circulating levels of LC-PUFAs compared to European Americans[15]. These large differences can be explained in part by variation in the allele frequencies of FADS variants associated with different biosynthetic efficiencies in these two populations[16]. Mathias et al. also revealed that African Americans have significantly higher levels of AA and lower levels of the AA precursor DGLA, and that FADS1 variants were significantly associated with AA, DGLA and the AA/DGLA ratio in a sample of fewer than 200 African Americans from the GeneSTAR study[15]. In addition, African ancestry populations have higher frequencies of the derived FADS haplogroup (represented by the variant rs174537 allele G)[17] that is associated with more efficient

conversion for PUFAs[16]. In contrast, Amerind ancestry Hispanic populations have higher frequencies of the ancestral FADS haplogroup (represented by rs174537 allele T) that has a reduced capacity to synthesize PUFAs. Accordingly, we demonstrated that higher global proportions of Amerind ancestry are associated with lower levels of PUFAs in Hispanic populations[17].

Genome-wide association studies (GWAS) of n-3 and n-6 PUFAs were performed by the CHARGE consortium in European ancestry (EUR) participants[18–20]. The CHARGE GWAS of n-3 PUFAs in 8,866 European Americans identified genetic variants in/near FADS1 and FADS2 associated with higher levels of ALA and lower levels of EPA and DPA, as well as SNPs in ELOVL2 associated with higher EPA and DPA and lower DHA. The CHARGE GWAS of n-6 PUFAs in 8631 European Americans confirmed that variants in the FADS gene cluster were associated with LA and AA, and it revealed that variants near NRBF2 were associated with LA and those in NTAN1 were associated with LA, GLA, DGLA, and AA (Fig. 1). In the Framingham Heart Offspring Study, variants in/near PCOLCE2, LPCAT3, DHRS4L2, CALN1 FADS1/2, and ELOVL2 were associated with PUFAs in European ancestry participants[21,22]. Collectively, these studies played an important role in identifying the genetic associations of n-3 and n-6 PUFAs in European ancestry populations.

To address the paucity of GWAS of PUFAs in non-European ancestry cohorts, we performed a meta-analysis of genome-wide association studies for n-3 and n-6 PUFAs for Hispanic American (HIS) and African American (AFA) participants from three CHARGE consortium cohorts: the Multi-Ethnic Study of Atherosclerosis (MESA), the Cardiovascular Health Study (CHS) and the Framingham Heart Study (FHS) Omni cohort. The major goals of the study were (1) to examine whether the major loci identified in European Americans are shared across race/ancestry groups, and (2) to examine evidence for genetic association unique to HIS and AFA populations. As GWAS approaches are not sufficient to identify the causal variants and determine the number of independent signals, especially in the context of long stretches of linkage disequilibrium (LD) within the FADS locus[15,23], we conducted statistical fine-mapping[24] to identify the most likely causal variants within each n-3 and n-6 PUFA-associated locus. We performed cross-ancestry replication analysis in CHARGE and MESA, with validation using the multi-ancestry GWAS of lipids from the Global Lipids Genetics Consortium (GLGC)[25]. Subsequently, we performed integrative analysis leveraging gene expression data from MESA[26,27] and the Genotype-Tissue Expression (GTEx) project[28] to identify genes that could contribute to our identified genetic association results. Finally, we examined open chromatin defined by ATAC-seq to determine the impact and physical contact of the identified variants with nearby genes (Fig. 2). Our study demonstrates the vital importance of diverse ancestry genetic studies for the study of complex traits, and particularly for metabolites that have been subject to evolutionary pressures and are closely regulated by specific protein-coding genes.

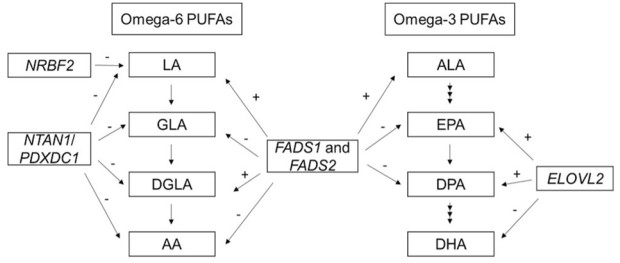

**Fig. 1 PUFAs metabolic pathway and summary of genome-wide association from previous CHARGE GWAS of n-3 and n-6 PUFAs in European Americans.** Figure 1 shows the summary of results from previous CHARGE GWAS of n-3 and n-6 PUFAs in European Americans. + and − signs indicate the direction of the associations for the minor allele of the most significant variant at each locus. The variants used to determine the directions of effect at each locus are as follows: FADS1 and FADS2: rs174547 (ALA, DPA, LA, GLA, DGLA, and AA); rs174538 (EPA). ELOVL2: rs780094 (DPA); rs3798713 (EPA); rs2236212 (DHA). NTAN1/PDXDC1: rs16966952 (LA, GLA, DGLA and AA). NRBF2: rs10740118 (LA).

## Results

**Participant characteristics**. The participants in the meta-analysis of GWAS for PUFAs included 1454 HIS and 2278 AFA-unrelated participants (Table 1; fatty acid levels are expressed as the percentage of total fatty acids throughout the entire manuscript). There were some differences in the distributions of fatty acid levels observed across cohorts, which were likely due to the sources of biospecimens for the assays (plasma phospholipids for MESA and CHS versus erythrocytes for FHS). For example, mean levels of DPA varied from 0.85% (CHS: plasma phospholipids) to

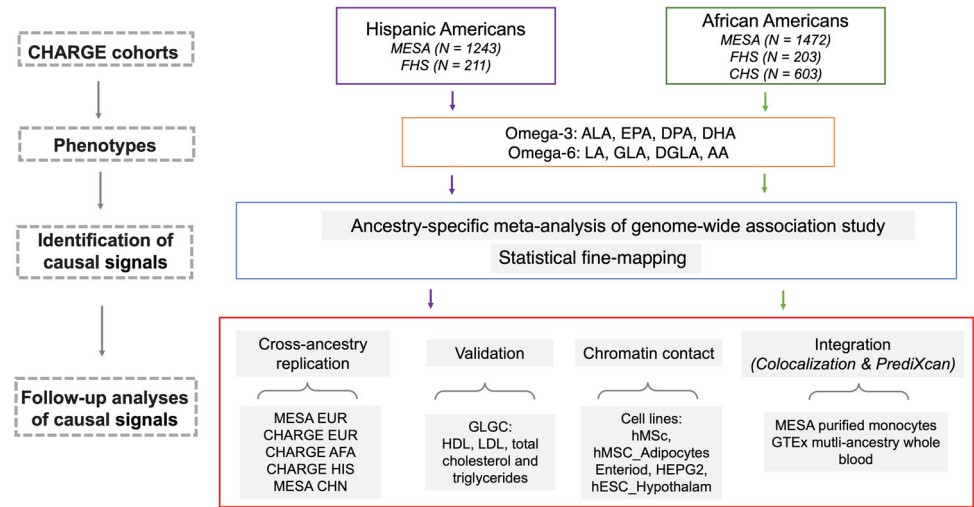

**Fig. 2 Study design.** Figure 2 shows the study design. GWAS of PUFAs was applied for each cohort stratified by HIS and AFA. Ancestry-specific GWAS meta-analysis and statistical fine-mapping were applied separately for HIS and AFA to identify the potential causal signals. Multiple follow-up analyses were conducted for the causal signals, including cross-ancestry replication, validation, chromatin contact analysis and integrative analyses.

### Table 1 CHARGE cohort descriptives.

| | MESA/Hispanic Americans | FHS/Hispanic Americans | MESA/African Americans | CHS/African Americans | FHS/African Americans |
|---|---|---|---|---|---|
| Participant characteristics | | | | | |
| No. subjects | 1243 | 211 | 1472 | 603 | 203 |
| Women | 629 (50.6) | 129 (61.1) | 788 (53.5) | 390 (64.7) | 130 (64.0) |
| Age, years | 61 [53, 69] | 53 [44, 60] | 63 [53, 70] | 74 [71, 79] | 58 [50, 67] |
| n-3 Polyunsaturated fatty acids | | | | | |
| ALA (% of total fatty acids) | 0.16 [0.12, 0.20] | 0.21 [0.16, 0.27] | 0.15 [0.12, 0.19] | 0.13 [0.11, 0.17] | 0.18 [0.15, 0.23] |
| EPA | 0.53 [0.37, 0.74] | 0.57 [0.47, 0.78] | 0.68 [0.51, 0.98] | 0.53 [0.39, 0.67] | 0.68 [0.48, 1.01] |
| DPA | 0.86 [0.73, 1.00] | 2.49 [2.13, 2.79] | 0.93 [0.80, 1.07] | 0.85 [0.75, 0.97] | 2.54 [2.25, 2.89] |
| DHA | 2.96 [2.29, 3.77] | 4.21 [3.45, 5.13] | 4.05 [3.25, 4.95] | 3.46 [2.87, 4.17] | 5.23 [4.21, 6.47] |
| n-6 Polyunsaturated fatty acids | | | | | |
| LA | 20.92 [18.87, 23.07] | 14.32 [12.24, 16.76] | 18.88 [17.12, 20.84] | 17.84 [16.46, 19.40] | 12.53 [10.88, 15.16] |
| GLA | 0.11 [0.08, 0.14] | 0.15 [0.10, 0.18] | 0.10 [0.08, 0.13] | 0.07 [0.05, 0.09] | 0.10 [0.07, 0.15] |
| DGLA | 3.57 [3.04, 4.13] | 1.95 [1.63, 2.35] | 2.89 [2.47, 3.33] | 2.76 [2.39, 3.24] | 1.51 [1.32, 1.78] |
| AA | 11.01 [9.37, 12.84] | 16.56 [15.17, 17.74] | 13.21 [11.65, 14.82] | 12.64 [11.57, 13.86] | 17.17 [15.95, 18.48] |

Table 1 shows the participant characteristics of the Hispanic Americans and African Americans from each cohort (MESA, CHS, and FHS). Data are presented as n (%) for binary measures or median [IQR] for continuous measures. Summary statistics are reported for the subset of individuals with data available for at least one of the fatty acid traits examined in genetic analyses. Fatty acids were measured in plasma phospholipids in MESA and CHS and in erythrocytes in FHS.

2.54% of total fatty acids (FHS: erythrocytes) in AFA and AA from 11.01% (MESA: plasma phospholipids) to 16.56% (FHS: erythrocytes) in HIS (Table 1). In addition, n-6 PUFAs, especially LA and AA, have relatively higher mean levels than n-3 PUFAs in all cohorts (Table 1).

Regardless of whether the fatty acids were measured in plasma phospholipids or erythrocytes, AFA populations had higher levels of AA and elevated ratios of AA to DGLA and AA to LA relative to Hispanic populations. This result would be expected given the frequency differences in the derived (efficient) to ancestral (inefficient) *FADS* haplogroups between these two populations. As expected, due to the lower levels of dietary ALA relative to LA entering the biosynthetic pathway, levels of n-3 LC-PUFAs including EPA, DPA, and DHA were significantly lower than the n-6 LC-PUFA, AA. In addition, African Americans had higher levels of n-3 LC-PUFAs than Hispanic Americans, again likely due to differences in the ratio of the derived to ancestral *FADS* haplogroups. These differences are similar to those observed examining the same PUFAs and LC-PUFAs and ratios when comparing African Americans and European Americans[15,29].

**Confirmation of top variants identified in prior CHARGE EUR GWAS of PUFAs.** We began by examining associations of seven known PUFA-associated signals from CHARGE EUR (summarized in Fig. 1) in our current study of CHARGE HIS and AFA. Multiple variants identified by previous CHARGE EUR GWAS meta-analyses[19,20] were also identified in CHARGE HIS (*FADS1/2* region: rs174547 and rs174538, *PDXDC1* variant: rs16966952 and *GCKR* variant: rs780094) and AFA (*FADS1/2* region: rs174547, *PDXDC1* variant: rs16966952, *GCKR* variant: rs780094 and *ELOVL2* variant: rs3734398) after adjusting for multiple testing for the number of variants examined across the eight PUFAs ($P < 0.05/8 = 0.006$) (Supplementary Data 1). The directions of effect observed in HIS and AFA for these variants were consistent with those reported for European ancestry populations in prior CHARGE GWAS meta-analyses of n-3 and n-6 PUFAs (Supplementary Data 1).

**GWAS and fine-mapping identify novel PUFA-associated genetic signals in CHARGE HIS and AFA.** Based on a

**Table 2 Genome-wide significant signals (Credible sets) for PUFAs in CHARGE Hispanic Americans.**

| | Lead variant (Chr:Pos:EFF:OTH) | EAF | Zscore | P-value | Cluster | # Of SNP | Novel/ Known | Nearest Gene |
|---|---|---|---|---|---|---|---|---|
| AA | rs102274 (11:61557826:C:T) | 0.506 | −24.26 | 5.1E−130 | 1 | 7 | Known | TMEM258 |
| | rs142068305 (11:67065755:T:G) | 0.196 | −7.06 | 1.63E−12 | 2 | 1 | Novel | ANKRD13D |
| | rs28364240 (11:67120530:G:C) | 0.204 | −7.04 | 1.88E−12 | 3 | 1 | Novel | POLD4 |
| | rs2668898 (11:61725498:G:A) | 0.402 | −5.83 | 5.32E−09 | 4 | 1 | Known | BEST1 |
| | rs180792704 (11:67325239:C:G) | 0.199 | −7.56 | 3.81E−14 | 5 | 1 | Novel | NA |
| | rs198434 (11:61483417:A:G) | 0.710 | −8.97 | 2.80E−19 | 6 | 1 | Novel | DAGLA |
| | rs518804 (11:57494487:C:A) | 0.420 | −7.73 | 1.01E−14 | 7 | 1 | Novel | TMX2 |
| | rs3177514 (11:66130358:G:T) | 0.699 | −5.60 | 2.06E−08 | 8 | 1 | Novel | SLC29A2 |
| ALA | rs174562 (11:61585144:G:A) | 0.503 | 7.84 | 4.30E−15 | 1 | 23 | Known | FADS1 |
| DGLA | rs174538 (11:61560081:A:G) | 0.488 | 14.70 | 6.03E−49 | 1 | 1 | Known | TMEM258 |
| | rs174585 (11:61611694:A:G) | 0.274 | 9.82 | 8.72E−23 | 2 | 1 | Known | FADS2 |
| | rs198434 (11:61483417:A:G) | 0.710 | 6.27 | 3.57E−10 | 3 | 1 | Novel | DAGLA |
| | rs198461 (11:61524366:C:A) | 0.363 | −5.95 | 2.54E−09 | 4 | 1 | Novel | MYRF |
| | rs57112407 (15:78088914:T:C) | 0.255 | −5.86 | 4.46E−09 | NA | NA | Novel | LINGO1 |
| | rs4985155 (16:15129459:G:A) | 0.524 | −7.72 | 1.16E−14 | 1 | 25 | Known | PDXDC1 |
| DPA | rs1535 (11:61597972:G:A) | 0.520 | −11.31 | 1.07E−29 | 1 | 18 | Known | FADS2 |
| | rs198434 (11:61483417:A:G) | 0.710 | −6.26 | 3.67E−10 | 2 | 1 | Novel | DAGLA |
| EPA | rs102274 (11: 61557826:C:T) | 0.506 | −11.56 | 6.18E−31 | 1 | 17 | Known | TMEM258 |
| GLA | rs174576 (11: 61603510:A:C) | 0.546 | −7.73 | 1.07E−14 | 1 | 19 | Known | FADS2 |
| LA | rs174564 (11:61588305:G:A) | 0.520 | 15.11 | 1.23E−51 | 1 | 10 | Known | FADS2 |
| | rs10751002 (11:63617634:G:T) | 0.664 | 6.06 | 1.36E−09 | 2 | 1 | Novel | MARK2 |
| | rs2668898 (11:61725498:G:A) | 0.402 | 5.54 | 2.99E−08 | 3 | 1 | Known | BEST1 |
| | rs28364240 (11:67120530:G:C) | 0.204 | 5.90 | 3.44E−09 | 4 | 1 | Novel | POLD4 |
| | rs11039018 (11:46909524:A:C) | 0.67 | −6.10 | 1.01E−09 | 5 | 1 | Novel | LRP4 |
| | rs518804 (11:57494487:C:A) | 0.420 | 6.03 | 1.62E−09 | 6 | 1 | Novel | TMX2 |

Table 2 shows the signals (credible sets) of putative causal variants identified for each of the PUFAs by fine- mapping using SuSiE in HIS (n = 1454). All variant positions are presented based on Human Genome Build 37. Variants previously documented in the CHARGE GWAS meta-analysis of n-3 and n-6 PUFAs were considered known prior to the current meta-analysis. The remaining variants were considered novel in the current study. There was only one genome-wide significant variant on chromosome 15 for DGLA (rs57112407) in HIS, and this signal was not carried forward for fine-mapping. P-values are calculated using a two-sided test for the z-score derived by meta-analysis including a total of n = 1454 biologically independent samples.

**Table 3 Genome-wide significant signals (Credible sets) for PUFAs in CHARGE African Americans.**

| | Lead variant (Chr:Pos:EFF:OTH) | EAF | Zscore | P-value | Cluster | # Of SNP | Novel/ Known | Nearest Gene |
|---|---|---|---|---|---|---|---|---|
| AA | rs174585 (11:61611694:A:G) | 0.060 | −9.32 | 1.08E−20 | 1 | 1 | Known | FADS2 |
| | rs174607 (11:61627321:C:G) | 0.078 | −6.49 | 8.47E−11 | 2 | 1 | Known | FADS2 |
| | rs174564 (11:61588305:G:A) | 0.133 | −14.85 | 6.43E−50 | 3 | 1 | Known | FADS2 |
| | rs174559 (11:61581656:A:G) | 0.078 | −13.68 | 1.27E−42 | 4 | 1 | Known | FADS1 |
| | rs17161592 (7:9388418:C:G) | 0.085 | −6.31 | 2.75E−10 | 1 | 2 | Novel | NA |
| DGLA | rs174560 (11:61581764:C:T) | 0.216 | 9.12 | 7.51E−20 | 1 | 1 | Known | FADS1 |
| | rs1136001 (16:15131974:T:G) | 0.220 | −6.11 | 9.69E−10 | 2 | 17 | Known | PDXDC1 |
| DPA | rs717894 (6:22119292:A:G) | 0.250 | −5.48 | 4.11E−08 | 1 | 1 | Novel | CASC15 |
| | rs9295741 (6:10997166:T:C) | 0.223 | 5.54 | 2.89E−08 | 2 | 2 | Known | ELOVL2 |
| DHA | rs114622288 (10:14663844:A:G) | 0.050 | −5.71 | 1.16e−08 | NA | NA | Novel | FAM107B |
| LA | rs1535 (11:61597972:G:A) | 0.163 | 7.88 | 3.14E−15 | 1 | 2 | Known | FADS2 |

Table 3 shows the signals (credible sets) of putative causal variants identified for each of the PUFAs by fine-mapping using SuSiE in AFA (n = 2278). All variant positions are presented based on Human Genome Build 37. Variants previously documented in the CHARGE GWAS meta-analysis of n-3 and n-6 PUFAs were considered known prior to the current meta-analysis. The remaining variants were considered novel in the current study. There was only one genome-wide significant variant on chromosome 10 for DHA (rs114622288) in AFA, and this signal was not carried forward for fine-mapping. P-values are calculated using a two-sided test for the z-score derived by meta-analysis including a total of n = 2278 biologically independent samples.

genome-wide significance threshold of $P < 5 \times 10^{-8}$, our complete GWAS of n-3 and n-6 PUFAs identified associations on chromosomes 11, 15 and 16 in CHARGE HIS (Table 2, Supplementary Fig. 1 and Supplementary Fig. 2) and chromosomes 6, 7, 10 and 11 in CHARGE AFA (Table 3, Supplementary Fig. 3 and Supplementary Fig. 4). For regions with more than one genome-wide significant variant, we applied statistical fine-mapping to identify the independent putative causal signals (credible sets) for

each genome-wide significant locus. We carried out these analyses separately for our CHARGE HIS and CHARGE AFA GWAS meta-analysis results.

We identified multiple independent putative causal signals for the PUFA traits [AA: 8 signals (credible sets); ALA: 1; DGLA: 5, DPA: 2; EPA: 1; GLA: 1; LA: 6] in HIS and [AA: 5; DGLA: 2, DPA: 2, LA: 1] in AFA (Tables 2, 3, Supplementary Data 2 and Supplementary Data 3). We examined the overlap of signals

identified from fine-mapping in HIS versus AFA. We observed that the credible sets were generally smaller in AFA (average number of variants in credible set: HIS:3.4; AFR:2.2) possibly driven by the lower average LD in AFA.

Among the independent credible sets identified, most were novel associated signals within a +/- 5 Mb region of the previously reported *FADS* signal on chromosome 11 (Tables 2, 3). Examining all the signals for PUFAs in HIS and AFA, we observed that the lead signal (reflecting the strongest evidence of association) on chromosome 11 represents the *FADS* signal reported in the previous GWAS[20]. For example, rs174547, the *FADS1* variant reported in the previous CHARGE EUR GWAS, is one of the variants in the first credible set for AA in HIS[19,20]. In addition to the known *FADS* signals, we also observed multiple novel independent signals at other regions of chromosome 11 for PUFAs [AA: 6 novel signals (credible sets) and LA: 3] in HIS, for example, in/near *ANKRD13D*, *TMX2*, *POLD4* and *SLC29A2* and spanning a long range (57.5 Mb ~ 67.1 Mb) on chromosome 11 for AA in HIS (Table 2). In addition, we observed several novel independent signals on other chromosomes showing associations with the PUFA traits in AFA [AA: 1 novel signal on chromosome 7 and DPA: 1 on chromosome 6] (Table 3).

**Additional independent PUFA-associated signals on chromosome 11 demonstrate chromatin contacts with *FADS* and other genes.** While prior studies have represented the *FADS* signal as primarily one signal[19,20], our study demonstrates numerous independent signals within the region (Table 2). For example, for AA we report signals intronic to *BEST1* and *DAGLA* within the *FADS* region (+/−1 Mb of the lead variant, rs102274; Fig. 3a). We examined this region to identify the subset of variants that may affect cis-regulatory elements in physical contact with nearby genes. Four variants within the credible sets in this region were located in regions of open chromatin defined by ATAC-seq and were in contact with gene promoters defined by Promoter Capture C in multiple metabolic-relevant cell types (human mesenchymal stem cells [hMSC], adipocytes derived from in vitro from the hMSC [hMSC_Adipocytes], induced pluripotent stem cell derived Hepatocytes [iPSC_Hepatocytes], embryonic stem cell derived Hypothalamic Neurons [hESC_HypothalamicNeurons], Enteroids, and HepG2s). Almost all of the interactions we detected were bait-to-bait interactions, meaning that they reflected physical contact between promoters of two different genes (Supplementary Data 4). For example, the region surrounding rs2668898 near *BEST1* showed evidence of physical contact with the *TMEM258*, *FADS1,* and *FADS2* region in multiple cell types and *TMEM258* region also showed evidence of physical contact with the *FADS1* and *FADS2* region (Fig. 4a and Supplementary Data 4). Besides the *FADS* region, we further found evidence of physical contact between *POLD4* and *ANKRD13D* (Fig. 4b and Supplementary Data 4), corresponding to the regions surrounding two signals identified in fine-mapping of AA in HIS (Fig. 3a).

**Novel signals on chromosome 11 identified in HIS show evidence of cross-ancestry replication or validation.** We investigated evidence of cross-ancestry replication for signals identified in our present GWAS of CHARGE HIS and AFA by examining evidence of genetic association in European Americans (CHARGE EUR[19,20] and MESA EUR), African Americans (CHARGE AFA), Hispanic Americans (CHARGE HIS) and Chinese Americans (MESA CHN). Replication analysis was performed with multiple testing correction (HIS: $P < 0.05/19$ signals $= 0.0026$ and AFA: $P < 0.05/11$ signals $= 0.004$; Supplementary Data 5 and Supplementary Data 6).

As noted previously, the first credible set identified in our present GWAS of HIS and AFA for each trait (reflecting the strongest evidence of association) generally coincided with the region of chromosome 11 reported in prior CHARGE GWAS efforts. These signals showed evidence of genetic association in European Americans, as well as across race/ancestry groups. For example, rs102274 for AA was replicated in the MESA EUR, CHARGE AFA, and MESA CHN groups (MESA EUR: $P = 1.04 \times 10^{-151}$, CHARGE AFA: $P = 2.36 \times 10^{-47}$, MESA CHN: $P = 8.75 \times 10^{-92}$) (Supplementary Data 5).

In addition, three novel signal were also replicated across race/ancestry groups (Table 4). Specifically, the *DAGLA* variant rs198434 and *MYRF* variant rs198461 in credible sets 3 and 4, respectively, for DGLA were replicated in analysis of MESA EUR (rs198434: $P = 2.54 \times 10^{-03}$ and rs198461: $P = 7.37 \times 10^{-09}$). *TMX2* variant rs518894 in credible set 6 for LA was replicated in CHARGE EUR ($P = 2.50 \times 10^{-03}$).

Some of the novel signals without cross-ancestry replication demonstrated large differences in allele frequencies across groups. For example, the effect allele frequency of rs28364240, a *POLD4* missense variant in credible set 3 for AA in Hispanics, is 0.204 in our CHARGE HIS group, but close to zero in the other race/ancestry groups examined (EUR: 0.003, AFR: 0.007, CHN: 0.005) (Fig. 3b, Supplementary Data 5, 7) and the effect allele frequency of rs142068305, a *ANKRD13D* intron variant, is 0.196 in our CHARGE HIS group while 0.007, 0.004, and 0.005 in AFR, EUR and CHN, respectively. These results suggest evidence of genetic association signals unique to HIS or other groups carrying Amerindian ancestry or admixture.

As some variants could not be interrogated using independent GWAS of PUFA traits, given those studies' focus on specific race/ancestry groups which may not include our variants of interest and/or limited sample sizes, we performed validation analyses using the results of multi-ancestry GWAS of lipid levels from the GLGC[25] including ~1.65 million individuals from five genetic ancestry groups (admixed African or African, East Asian, European, Hispanic and South Asian). We focused on the most significant putative causal variants from each credible set and applied multiple testing correction for the number of validated variants (HIS: $P < 0.05/19 = 0.0026$ and AFA: $P < 0.05/11 = 0.004$). Interestingly, we observed that multiple novel signals without cross-ancestry replication did demonstrate association with one or more lipid levels. For example, the LA associated *LRP4* variant rs11039018 was validated based on its association with HDL and Triglycerides (HDL: $P = 2.85 \times 10^{-74}$ and Triglycerides: $P = 4.50 \times 10^{-43}$), while the LA associated *MARK2* intron variant rs10751002 was validated based on its association with LDL and Total Cholesterol (LDL: $P = 3.31 \times 10^{-12}$ and Total Cholesterol: $P = 5.74 \times 10^{-09}$) (Table 4, Supplementary Data 8 and Supplementary Data 9).

**Integrative analyses identify putative causal genes and pathways for the PUFA loci.** Using colocalization with eQTL resources, we identified candidate genes underlying the genetic association signals for the PUFA traits. In HIS, we found colocalization with expression of the genes *MED19, TMEM258, PACS1, RAD9A, C11orf24, CTTN* on chromosome 11 and *PDXDC1* on chromosome 16 based on MESA multi-ancestry eQTL resources[26] (Table 5 and Supplementary Data 10). In further analysis using eQTL resources from GTEx whole blood[28], we confirmed colocalization with *TMEM258* and *MED19* identified using the MESA multi-ancestry eQTLs, and also identified colocalization with *FADS1, RPS4XP13, AP001462.2, PGA5, PGA5, TPCN2, MEN1* on chromosome 11 and *RP11-156C22.5* on chromosome 16. (Table 5 and Supplementary Data 11).

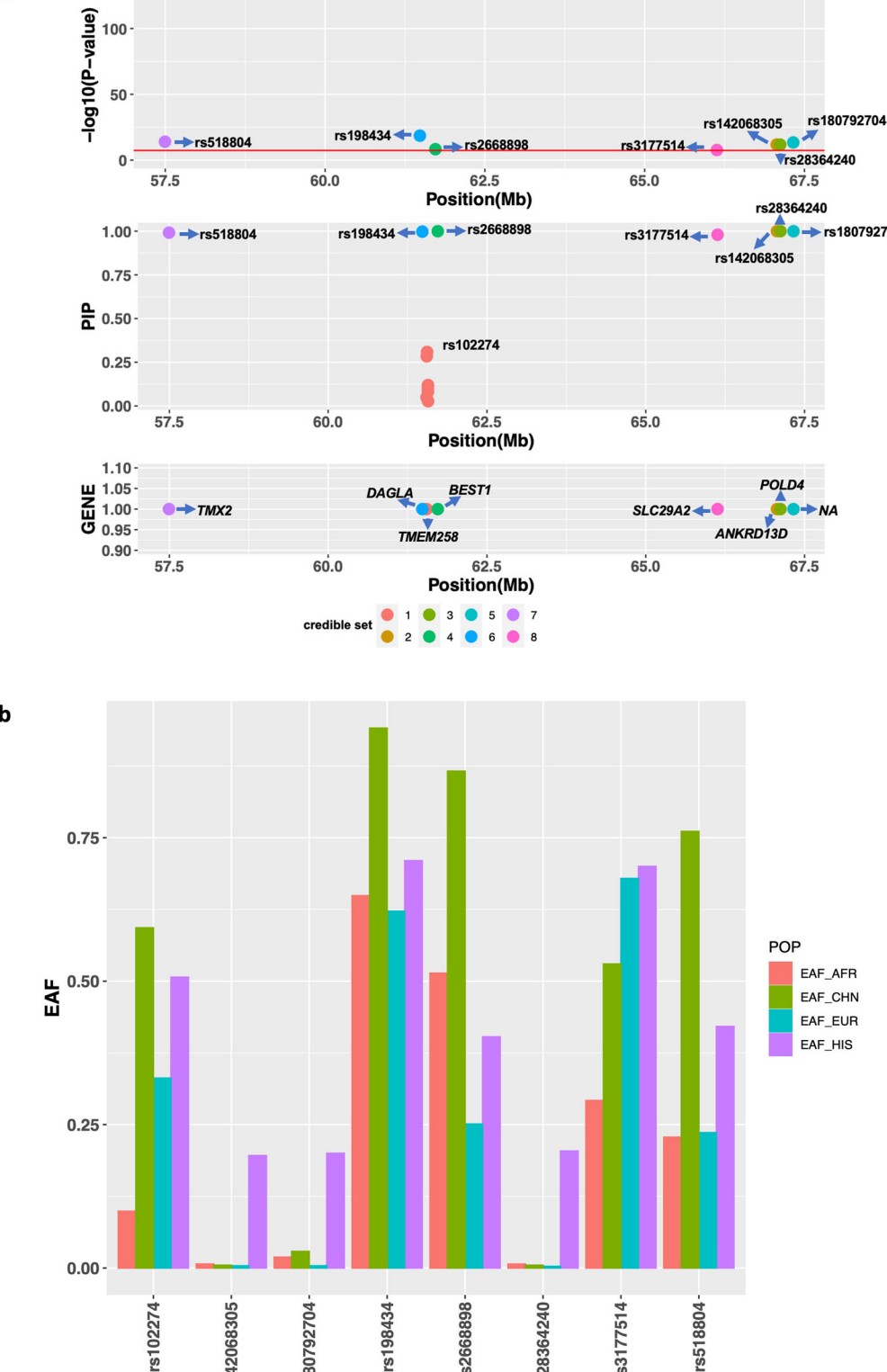

**Fig. 3 Summary of signals (credible sets) identified in association with AA on chromosome 11 in Hispanic Americans.** Panel (**a**) shows detailed information for the identified signals. The upper display shows the *P*-value of the putative causal variants of each signal (credible set) on chromosome 11 from GWAS based on data for a total of *n* = 1454 biologically independent samples; middle display shows the Posterior Inclusion Probability (PIP) of the putative causal variants from statistical fine-mapping using SuSIE; bottom display shows the Gene near/in the putative causal variants of each signal. Panel (**b**) shows the effect allele frequencies (EAF) in MESA across four self-reported race/ethnic groups (African American [*n* = 2278], Chinese [*n* = 648], Hispanic American [*n* = 1454], and European ancestry [*n* = 2344]) for the most significant putative causal variant from each signal (credible set). Source data for the figure are provided in Supplementary Data 7.

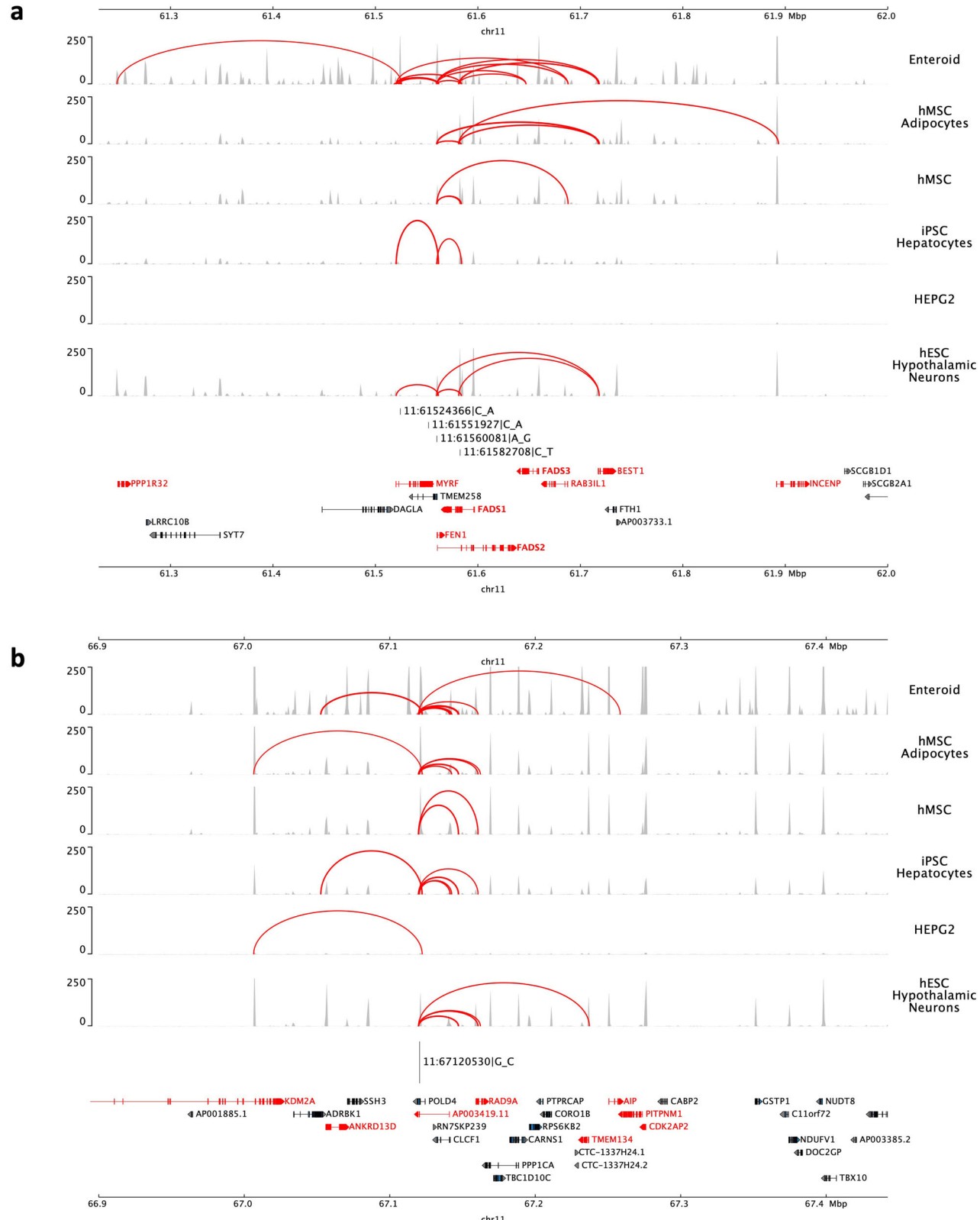

**Fig. 4 Chromatin contact analysis of selected genome-wide significant variants identified on Chromosome 11.** Figure 4 shows the chromatin contacts for the putative causal variants within the selected signals (**a**: *FADS* region and **b:** *POLD4* region) located in open chromatin defined by ATAC-seq with gene promoters defined by Promoter Capture C (implicated genes highlighted in red) in multiple metabolic-relevant cell types. The cell types examined include: human mesenchymal stem cells (hMSC), which were also differentiated in vitro to adipocytes (hMSC_Adipocytes), induced pluripotent stem cell derived Hepatocytes (iPSC_Hepatocytes), embryonic stem cell-derived Hypothalamic Neurons (hESC_HypothalamicNeurons), Enteroids, and HepG2s. The *y* axis shows the ATAC-seq read density normalized using the reads per genomic content (RPGC) method. All variant positions are presented based on Human Genome Build 37.

**Table 4 Novel PUFA-associated signals (credible sets) from analysis of HIS with external cross-ancestry replication or multi-ancestry validation evidence.**

| Traits | Variants (chr:pos:effect:other) | Discovery | Replication | Validation | Direction | Nearest Gene |
|---|---|---|---|---|---|---|
| AA | rs518804 (11:57494487:C:A) | HIS: $P=1.01E-14$ | NS | HDL: $P=1.96E-06$<br>logTG: $P=0.001$ | HIS: (−)<br>HDL: (−)<br>logTG: (+) | *TMX2* |
|  | rs198434 (11:61483417:A:G) | HIS: $P=2.80E-19$ | NS | logTG: $P=1.65E-03$ | HIS: (−)<br>logTG: (+) | *DAGLA* |
| DGLA | rs198461 (11:61524366:C:A) | HIS: $P=2.54E-09$ | EUR: $P=7.37E-09$ | HDL: $P=4.81E-13$<br>LDL: $P=1.92E-13$<br>logTG: $P=1.19E-18$<br>TC: $P=5.63E-14$ | HIS: (−)<br>EUR: (−)<br>HDL: (+)<br>LDL: (+)<br>logTG: (−)<br>TC: (+) | *MYRF* |
|  | rs198434 (11:61483417:A:G) | HIS: $P=3.57E-10$ | EUR: $P=2.54E-03$ | logTG: $P=1.65E-03$ | HIS: (+)<br>EUR: (+)<br>logTG: (+) | *DAGLA* |
| DPA | rs198434 (11:61483417:A:G) | HIS: $P=3.67E-10$ | NS | logTG: $P=1.65E-03$ | HIS: (−)<br>logTG: (+) | *DAGLA* |
| LA | rs518804 (11:57494487:C:A) | HIS: $P=1.62E-09$ | EUR: $P=2.50E-03$ | HDL: $P=1.96E-06$<br>logTG: $P=0.001$ | HIS: (+)<br>EUR: (−)<br>HDL: (−)<br>logTG: (+) | *TMX2* |
|  | rs10751002 (11:63617634:G:T) | HIS: $P=1.36E-09$ | NS | LDL: $P=3.31E-12$<br>TC: $P=5.74E-09$ | HIS: (+)<br>LDL: (+)<br>TC: (+) | *MARK2* |
|  | rs1039018 (11:46909524:A:C) | HIS: $P=1.01E-09$ | NS | HDL: $P=2.85E-74$<br>logTG: $P=4.5E-43$ | HIS: (+)<br>HDL: (+)<br>logTG: (−) | *LRP4* |

Table 4 shows the novel putative causal variants in each signal (credible set) identified from fine-mapping for PUFAs with replication and validation evidence in HIS ($n=1454$). All variant positions are presented based on Human Genome Build 37. Variants that were not previously documented in the CHARGE GWAS meta-analysis of n-3 and n-6 PUFAs and were not in LD with known GWAS variants were considered novel in the current study. *P*-values corresponding to discovery (in HIS) and replication (in EUR) are calculated using a two-sided test for the z-score derived by meta-analysis including a total of $n=1454$ or $n=2344$ biologically independent samples, respectively. Validation *P*-values are extracted directly from the GWAS summary statistics corresponding to the GLGC publication[25].

We also performed complementary integrative analysis using PrediXcan, identifying significant associations for predicted expression of *TMEM258* with AA, ALA, DGLA, DPA, EPA, GLA, and LA (after multiple testing correction for all genes examined: $P < 0.05/4470 = 0.00001$), based on integration with eQTL from both MESA and GTEx. PrediXcan also identified *TMEM109, ZBTB3, TTC9C, POLD4, INCENP,* and *FERMT3* on chromosome 11 and *PDXDC1* on chromosome 16 as putative genes associated with PUFAs in HIS (Table 5, Supplementary Data 12 and Supplementary Data 13). For AFA, colocalization and PrediXcan analyses did not identify any genes of interest that met our pre-specified thresholds for statistical significance.

Incorporating the prior chromatin contacts identified (Supplementary Data 4), we found that several of our GWAS regions had physical contact with one or more genes identified by integration with eQTL resources. For example, *RAD9A* was supported by colocalization with MESA eQTL[26] and also showed chromatin contact with *POLD4* in nearly all cell types examined (Fig. 4b). In addition, *INCENP* was supported by PrediXcan using both MESA[26] and GTEx[30] resources and also showed chromatin contact with *TMEM258, FADS1,* and *FADS2* in nearly all cell types examined (Fig. 4a). We further observed that *CLCF1, RAD9A, FADS2, TMEM258, INCENP, FADS1* identified from colocalization or PrediXcan were additionally supported by chromatin contact analyses (Fig. 4, Supplementary Data 4).

To follow-up on the genes of interest identified by colocalization and PrediXcan analyses, we examined their co-expression with *FADS1* using GTEx whole blood gene expression[28] with multiple testing correction for the number of genes under consideration (HIS: $P < 0.05/39 = 0.0012$). In both unadjusted and age/sex-adjusted regression models, multiple genes showed statistically significant co-expression with *FADS1*, for example, *TMEM258, MED19, POLD4, RAD9A,* and *SSH3* (Supplementary Data 14), suggesting these genes have shared patterns of expression.

We further applied gene set enrichment analysis to the set of genes identified by our integrative colocalization and PrediXcan analyses using the Molecular Signatures Database (MSigDB)[31–33] gene sets (Supplementary Data 15). The most significantly enriched gene set (NIKOLSKY_BREAST_CANCER_11Q12_-Q14_AMPLICON) comprised the set of genes within amplicon 11q12-q14 identified in a copy number alterations study of 191 breast tumor samples[34] ($P = 6.71 \times 10^{-17}$), which included twelve genes from among those identified by the integrative follow-up analyses of our GWAS results: *RAD9A, CTTN, PGA5, TPCN2, TMEM109, POLD4, CLCF1, SSH3, TBC1D10C, CCS, BBS1,* and *DPP3.* The second most significantly enriched gene set (PEA3_Q6) represents the set of genes having at least one occurrence of the motif ACWTCCK in the regions spanning 4 kb centered on their transcription starting sites ($P = 3.25 \times 10^{-09}$), which included eight genes from among those identified in our integrative analyses: *TMEM258, C11orf24, FERMT3, POLD4, TBC1D10C, CCDC88B, MAP4K2,* and *DPP3.*

## Discussion

To address the relative lack of prior studies examining the genetics of PUFA levels in non-European ancestry populations, we carried out a meta-analysis of GWAS of n-3 and n-6 PUFAs in HIS and AFA across three cohorts: MESA, CHS and FHS. Examining genetic variants identified in prior CHARGE GWAS of the same traits in European Americans[19,20], we demonstrated evidence of association with n-3 and n-6 PUFAs for the signals in/near *FADS1/2* on chromosome 11, *PDXDC1* on chromosome 16, and *GCKR* on chromosome 2 in both HIS and AFA from our current CHARGE GWAS, as well as for *ELOVL2* on chromosome 6 in AFA only.

Through genome-wide analysis and subsequent statistical fine-mapping of our ancestry-specific results, we demonstrated evidence of multiple independent novel signals within the *FADS1/2*

**Table 5 Integrative analysis (Colocalization and PrediXcan) in the Hispanic Americans using multi-ancestry resources from MESA and GTEx.**

| | Colocalization Analysis | | PrediXcan | |
|---|---|---|---|---|
| | MESA multi-ancestry eQTLs | GTEx eQTLs | MESA | GTEx |
| AA | Chromosome 11 MED19, TMEM258, PACS1, RAD9A | RPS4XP13, AP001462.6 | TMEM258, TMEM109, ZBTB3, TTC9C, FERMT3, MED19, POLD4, CLCF1, INCENP, MADD, SSH3, C11orf24, PRPF19, TBC1D10C, BANF1, CCDC86, NXF1, MS4A6E, CCS, COX8A, CCDC88B, ACP2, MAP4K2 | TMEM258, TMEM223, NXF1, INCENP, MUS81, C11orf84, MED19, MEN1, BBS1, NEAT1, DPP3, SSH3, ACP2, ASRGL1, RNASEH2C |
| ALA | Chromosome 11 TMEM258, MED19 | MED19, PGA5, TMEM258 | TMEM258, TMEM109 | TMEM258 |
| DGLA | Chromosome 11 TMEM258; Chromosome 16 PDXDC1 | RP11-426C22.5 | TMEM258, ZBTB3; PDXDC1 | TMEM258, FADS1, FADS2 |
| DPA | Chromosome 11 TMEM258, C11orf24, RAD9A | PGA5 | TMEM258, TMEM109 | NPIPA2 |
| EPA | Chromosome 11 TMEM258 | TPCN2 | TMEM258, FERMT3, TMEM109 | TMEM258, SSH3, TMEM223 |
| GLA | Chromosome 11 TMEM258 | MEN1 | TMEM258 | TMEM258 |
| LA | Chromosome 11 MED19, CTTN, C11orf24, RAD9A | MED19, TPCN2, FADS1, RPS4XP13, AP001462.6 | TMEM258, TMEM109, FERMT3, ZBTB3, COX8A, MADD, POLD4, TBC1D10C, INCENP, TTC9C, MED19, CLCF1, SSH3, ACP2 | TMEM258, INCENP, SSH3, C11orf84, TMEM223, GIF, NXF1, MED19, MUS81, ACP2 |

Table 5 shows the results of integrative analysis including colocalization analysis and PrediXcan in HIS by using MESA and GTEx eQTL data. For colocalization analysis, eQTL resources include MESA multi-ancestry eQTL from purified monocytes and GTEx European ancestry whole blood tissue eQTL. GWAS signals with posterior colocalization probability of hypothesis 4 (PP.H4) > 0.80, or PP.H4 > 0.50 and the ratio of PP.H4 / PP.H3 > 5 were considered colocalized with eQTL. For PrediXcan, reference gene expression prediction models include MESA purified monocytes and GTEx European ancestry whole blood, and multiple testing correction was applied across all genes tested (MESA: $P < 0.05/4470 = 0.00001$ and GTEx: $P < 0.05/4350 = 0.00001$).

locus in both HIS and AFA, and in/near *ELOVL2* in AFA. Among these independent novel signals, we found three signals identified in HIS demonstrated evidence of replication in AFA based on association with the same PUFA traits in MESA and CHARGE (LA: rs518804 intronic to *TMX2* [Thioredoxin related transmembrane protein 2]; DGLA: rs198461 intronic to *MYRF* [Myelin regulatory factor] and rs198434 intronic to *DAGLA* [Diacylglycerol lipase alpha]). In addition, multiple novel signals without cross-ancestry replication did show evidence of validation based on association with lipid levels in GLGC[25]. For example, rs11039018, a *LRP4* (LDL receptor-related protein) intron variant associated with AA and LA was validated based on its association with HDL and Triglycerides. This finding is supported by animal studies showing that deficiency of *Lrp4* in adipocytes increased glucose and insulin tolerance and reduced serum fatty acids[35]. Prior studies from the FORCE consortium have shown that LA is associated with lower risk of diabetes, thus it is possible that the association of *LRP4* on diabetes risk factors is mediated by LA[36]. In addition, a *MARK2* (microtubule affinity regulating kinase 2) intron variant rs10751002 associated with LA was validated based on its association with LDL and total cholesterol. We chose to perform validation analysis using association results for lipid levels from the GLGC[25] due to (1) the large sample size (>1 million), which made our validation effort very well powered to detect associations with the selected lipid traits, and (2) the association between fatty acids and lipid traits, for example, fish oil supplements lowering triglycerides[37] and dietary linoleic acid lowering cholesterol[38,39].

While we identified three signals in HIS with evidence of cross-ancestry replication, we also found a large number of signals in HIS that could not be replicated across race/ancestry groups (European, African American and Chinese), in part to differences in allele frequencies. For example, the chromosome 11 *POLD4* (DNA polymerase delta 4, accessory subunit) missense variant rs28364240 and *ANKRD13D* (ankyrin repeat domain 13D) intron variant rs142068305 identified in association with AA have minor allele frequencies of 0.204 and 0.196 in HIS, compared to frequencies close to zero in other race/ancestry groups.

Examining the distance between the putative causal variants in different credible sets identified in HIS, we observed that the signals on chromosome 11 span a long range (57.5 Mb ~ 67.1 Mb). The extended physical distance covered by these independent PUFA-associated variants, combined with their subsequent validation in association with selected lipid traits, suggests there may be long-range chromatin interactions or other forms of physical interaction that may have yielded distinct genetic associations across this region[40]. Interestingly, prior studies have reported the *FADS* signal on chromosome 11 as primarily just one genetic signal[19,20]. However, our study provides evidence of two more independent signals (*BEST1* and *DAGLA*) within this *FADS* region. In order to understand the chromatin interactions of the *FADS* region on chromosome 11, we used ATAC-seq peaks and chromatin loops to perform the chromatin contact analyses. We identified multiple genes from colocalization or PrediXcan also supported by chromatin contacts, including *CLCF1*, *RAD9A*, *FADS2*, *TMEM258*, *INCENP* and *FADS1*, providing support for the role of our identified genetic signals in regulating these genes. In addition, we observed evidence of chromatin contacts among multiple distinct credible sets identified based on our fine-mapping of genetic signals on chromosome 11. For example, the region surrounding rs2668898 near *BEST1* also showed evidence of physical contact with the *TMEM258*, *FADS1*, and *FADS2* region in multiple cell types and *TMEM258* also showed evidence of physical contact with the *FADS1* and *FADS2* region. This support for physical contact among some of the multiple independent signals within the *FADS* region opens

the possibility of coordinated regulation among these distinct genetic signals. Besides the *FADS* region, *POLD4* also showed evidence of physical contact with the *ANKRD13D* region in multiple cell types. The cell types examined for chromatin interaction correspond to pancreas, liver, and other cell types that could play a role in synthesis and regulation of fatty acids. While the cell types used to examine chromatin interactions are distinct from those used for our integrative eQTL analyses, the chromatin interaction results do provide support for the plausible role of the genes identified by colocalization and PrediXcan.

Through integrative analyses, including colocalization analysis and PrediXcan, that examined overlap of our GWAS of PUFA levels with selected eQTL resources[26,28], we identified putative candidate genes that may shed light on the functional mechanisms of our identified genetic association signals. On chromosome 11 containing the *FADS* genes, we identified overlap with eQTL for multiple other genes including *MED19* (Mediator Complex Subunit 19), *TMEM258* (Transmembrane Protein 258), *PACS1* (Phosphofurin Acidic Cluster Sorting Protein 1), *RAD9A* (RAD9 Checkpoint Clamp Component A) and *CTTN* (Cortactin) suggesting additional complexity within this region beyond the *FADS* genes. For the signals on chromosome 16 identified based on analyses of DGLA in HIS and AFA, in/near *NTAN1* and *PDXDC1*, our integrative PrediXcan analyses identified *PDXDC1* (Pyridoxal Dependent Decarboxylase Domain Containing 1) (but not *NTAN1*) as a putative gene for DGLA. In addition, having identified association with AA in HIS for the *POLD4* missense variant rs28364240, our subsequent identification of *POLD4* (DNA Polymerase Delta 4, Accessory Subunit) based on the PrediXcan analyses brings additional support for this gene. To follow-up on the genes of interest identified by colocalization and PrediXcan analyses, we examined their co-expression with *FADS1* using GTEx whole-blood gene expression. Multiple genes on chromosome 11 identified in our integrative analyses combining the GWAS of PUFAs with whole blood expression from GTEx showed evidence of co-expression with *FADS1*, for example, *TMEM258*, *POLD4*, *TMEM109*, and *ZBTB3*. This finding suggests some genomic regions at a considerable distance from *FADS1* may play a role in regulating its expression, and ultimately influence circulating PUFA levels. Downstream pathway analysis of the genes identified by our integrative analyses further highlighted common features of these genes, including their regulation by transcription factors[41] and their relevance to breast cancer[34], UV radiation[42], and cell states or perturbations within the immune system[43,44]. As a recent Mendelian randomization study highlighted the relationship between genetically elevated PUFA levels and risk of cancer[45], our current work provides further support for that connection.

While our genetic association study of PUFA levels in HIS and AFA provides novel insights, our work has several limitations. First, while we have combined data from multiple CHARGE cohorts, the overall sample size of our study is still relatively small for a GWAS. Second, as we began this GWAS effort some years ago, our work makes use of older imputation panels based on the 1000 Genomes. We expect future work could leverage newer resources including imputation based on the Trans-omics for Precision Medicine (TOPMed) reference panel or newer whole genome sequence data from TOPMed[46]. Third, the circulating PUFA levels examined in this study are derived from heterogeneous sources (plasma phospholipids in MESA and CHS vs. erythrocytes in FHS), which could have resulted in heterogeneity of genetic associations across the included studies and overall loss of power. Finally, while our integration of GWAS with eQTL proved useful in some cases, our efforts were driven in part by the available resources. We made use of multi-ancestry eQTL resources based on purified monocytes in MESA, as we knew

these resources were well-matched with our GWAS cohorts in terms of LD structure, although purified monocytes were likely not the most relevant cell type for our study. We complemented those efforts with whole blood eQTL from GTEx through which we were able to capture colocalization of *FADS1* that was not observed in MESA due to the lack of significant cis-eQTL for *FADS1*. This limitation underscores the need for more diverse ancestry eQTL resources across a wider range of tissues and cell types.

In summary, working with the CHARGE Consortium, we conducted a consortium-based GWAS of circulating PUFA levels in HIS and AFA cohorts. Our study demonstrated evidence of shared genetic influences on PUFA levels across race/ancestry groups, and demonstrated a large number of distinct genetic association signals within a neighborhood of the well-documented *FADS* region on chromosome 11[19,20]. Our findings provide insight into the complex genetics of circulating PUFA levels that reflect, in part, their response to evolutionary pressures across the course of human history[47,48]. Overall, our study demonstrates the value of investigating complex trait genetics in diverse ancestry populations and highlights the need for continued efforts for expanded genetic association efforts in cohorts with genetic ancestry that reflects that of the general population within the United States and worldwide. In future work, genetic loci identified in this study could be leveraged to examine gene x fatty acid interactions on disease outcomes, or to construct more precise genetic predictors of sub-optimal or deficient fatty acid levels, which could be central to efforts in precision nutrition[17,49]. In addition, we anticipate the results from this work could help to motivate downstream studies focused on fatty acids as a mediator of specific genes' influences on identified pathways, including cancer and immune responses, as well as the long-range regulation of gene function by other genes located in distinct and distant portions of the same chromosome.

## Methods

**Study participants**. The participants in this study were recruited from three population-based cohorts: the Multi-Ethnic Study of Atherosclerosis (MESA), the Cardiovascular Health Study (CHS), and the Framingham Heart Study (FHS). This manuscript focuses on HIS participants from MESA ($N = 1243$) and FHS ($N = 211$) and AFA participants from MESA ($N = 1472$), CHS ($N = 603$), and FHS ($N = 203$).

MESA is a longitudinal cohort study of subclinical cardiovascular disease and risk factors that predict progression to clinically overt cardiovascular disease or progression of subclinical disease[50]. Between 2000 and 2002, MESA recruited 6814 men and women 45–84 years of age from Forsyth County, North Carolina; New York City; Baltimore; St. Paul, Minnesota; Chicago; and Los Angeles. Participants at baseline were 38% White, 28% African American, 22% Hispanic, and 12% Asian (primarily Chinese) ancestry.

CHS is a population-based cohort study of risk factors for coronary heart disease and stroke in adults ≥65 years conducted across four field centers[51]. The original predominantly European ancestry cohort of 5201 persons was recruited in 1989–1990 from random samples of the Medicare eligibility lists; subsequently, an additional predominantly African-American cohort of 687 persons was enrolled in 1992–1993 for a total sample of 5888. Analyses were limited to those with available DNA who consented to genetic studies.

FHS is a population-based longitudinal study of families living in Framingham, Massachusetts which originated in 1948 and consisted of individuals of predominantly European ancestry[52]. In 1994, the Omni Cohort 1 enrolled 507 men and women of African-American, Hispanic, Asian, Indian, Pacific Islander and Native American origins, who at the time of enrollment were residents of Framingham and the surrounding towns.

**Fatty acid measurements**. Circulating PUFA levels were quantified from plasma phospholipids in MESA and CHS, and from erythrocytes in FHS. Measurements were taken from biologically independent distinct samples.

*MESA*. The fatty acids were measured in EDTA plasma, frozen at –70 °C[53]. Lipids were extracted from the plasma using a chloroform/methanol extraction method and the cholesterol esters, triglyceride, phospholipids and free fatty acids are separated by thin layer chromatography. The fatty acid methyl esters were obtained from the phospholipids and were detected by gas chromatography flame

ionization. Individual fatty acids were expressed as a percent of total fatty acids. A total of 28 fatty acids were identified.

*CHS*. Blood was drawn after a 12-h fast and stored at –70 °C. Total lipids were extracted from plasma using methods of Folch[54], and phospholipids separated from neutral lipids by one-dimensional TLC. Fatty-acid-methyl-ester (FAME) samples were prepared by direct transesterification using methods of Lepage and Roy[55], and separated using gas chromatography (Agilent5890 gas- chromatograph-FID-detector; Supelco fused-silica 100 m capillary column SP-2560; initial 160 °C 16 min, ramp 3.0 °C/min to 240 °C, hold 15 min)[56]. Identification, precision, and accuracy were continuously evaluated using model mixtures of known FAMEs and established in-house controls, with identification confirmed by GC-MS at USDA (Peoria, IL). A total of 42 fatty acids were identified. Fatty acid levels were expressed as percent of total fatty acids. CVs were <3% for most fatty acids.

*FHS*. Red blood cells (RBCs) were isolated from blood drawn after a 10–12 h fast and frozen at −80 °C immediately after collection. RBC fatty acid composition was analyzed by gas chromatography (GC) with flame ionization detection[57]. Unwashed, packed RBCs were directly methylated with boron trifluoride and hexane at 100 °C for 10 min. The fatty acid methyl esters thus generated were analyzed using a GC2010 Gas Chromatograph (Shimadzu Corporation, Columbia, MD) equipped with an SP2560, fused silica capillary column (Supelco, Bellefonte, PA). Fatty acids were identified by comparison with a standard mixture of fatty acids characteristic of RBC (GLC 727, NuCheck Prep, Elysian, MN) which was also used to determine individual fatty acid response factors. The omega-3 index is defined as the sum of EPA and DHA expressed as a percent of total identified fatty acids. The coefficients of variation were 6.2% for EPA, 4.4% for DHA and 3.2% for the omega-3 index. All fatty acids present at >1% abundance had CVs of ≤7%.

**Genotyping and imputation**. Each of the participating cohorts had genome-wide genotype data based on a GWAS array, followed by imputation based on the 1000 Genomes Phase 1 v3 (for CHS) or Phase 3 (for MESA and FHS) Cosmopolitan reference panel[58].

*MESA*. Participants in the MESA cohort who consented to genetic analyses and data sharing (dbGaP) were genotyped using the Affymetrix Human SNP Array 6.0 (GWAS array) as part of the NHLBI CARe (Candidate gene Association Resource) and SHARe (SNP Health Association Resource) projects. Genotype quality control for these data included filter on SNP level call rate < 95%, individual level call rate < 95%, heterozygosity > 53%[59]. The cleaned genotypic data was deposited with MESA phenotypic data into dbGaP (study accession phs000209.v13.p3); 8224 consenting individuals (2685 White, 2588 non-Hispanic African-American, 2174 Hispanic, 777 Chinese) were included, with 897,981 SNPs passing study specific quality control (QC). SNP coverage from the original GWAS SNP genotyping array was increased through imputation using the 1000 Genomes Phase 3 integrated variant set completed using the Michigan Imputation Server[60,61].

*CHS*. DNA was extracted from blood samples drawn on all participants at their baseline examination. In 2010, genotyping was performed at the General Clinical Research Center's Phenotyping/Genotyping Laboratory at Cedars-Sinai using the Illumina HumanOmni1-Quad_v1 BeadChip system on African-American CHS participants who consented to genetic testing, and had DNA available for genotyping. Genotyping was attempted in 844 participants, and was successful in 823. Participants were excluded if they had a call rate <= 95% or if their genotype was discordant with known sex or prior genotyping (to identify possible sample swaps). Genotype quality control excluded SNPs with a call rate < 97%, HWE $P < 1 \times 10^{-5}$, >1 duplicate error or Mendelian inconsistency (for reference CEPH trios), heterozygote frequency = 0, which resulted in a final set of 963,248 SNPs (940,567 autosomal). Imputation to the 1000 Genomes Phase I integrated variant set was completed using IMPUTE version 2.2.2. Variants with insufficient effective minor alleles are filtered prior to analysis, with a threshold set at 5 effective alleles resulting in 14,191,388 variants for analysis.

*FHS*. Direct genotypes were obtained using the Affymetrix 500 K and MIPS 50 K chips, and were analyzed at the Affymetrix Core Laboratory. Genotype quality control for these data included filter on SNP level call rate < 95%, individual level call rate < 95%, HWE P < 10-5, and genotypes with Mendel errors were set to missing. The cleaned genotypic data consisted of $N = 414$ (211 Hispanic, 203 African-American) with 628,076 SNPs passing study specific quality control (QC). SNP coverage from the original GWAS SNP genotyping array was increased through imputation using the 1000 Genomes Phase 3 integrated variant set completed using the Michigan Imputation Server[60,61].

**Data transformation and detection of outliers in measured PUFA levels**. After examining the raw phenotype distributions for each of the phenotypes of interest, we applied variable transform for traits exhibiting deviation from normality. Log-transformation was applied for ALA, EPA, and GLA. In addition, outliers for all of the PUFA levels were identified by the limits of median +/- 3.5 * MAD', where MAD' is computed with a scale factor constant of 1.4826 [default for the mad()

function in R]. The value of MAD' $= 1.4826 *$ MAD0 where MAD0 is the raw value of median absolute deviation (MAD). For all the PUFAs, outliers were winsorized to the value of (median $+/- 3.5 *$ MAD').

**Genome-wide association study (GWAS) and meta-analysis.** Participants who were not in the self-reported African American or Hispanic American groups of interest to this manuscript were excluded from the primary GWAS analyses. To construct clean race/ancestry groups for stratified GWAS analyses, self-reported race/ethnicity groups were cleaned by removing outliers for principal components (PCs) of ancestry based on the limited of mean $+/- 3.5 *$ standard deviation, separately for each of the participating cohorts. GWAS was then carried out separately in each cohort and stratified by race/ancestry with covariate adjustment for age, sex, study site, and PCs of ancestry. Cohort-specific GWAS results were filtered using EasyQC based on minor allele count (MAC) $> 6$ and imputation R-squared $> 0.3$. Cohort-specific results were combined using weighted sum of z-score meta-analysis in METAL[62] and filtered using Effective Heterozygosity Filter (effHET) $> 60$. A threshold of $P < 5 \times 10^{-8}$ was applied to identify genome-wide significant loci.

**Statistical fine-mapping using SuSiE.** For each chromosome with more than one genome-wide significant variant (at $P < 5 \times 10^{-8}$), we carried out statistical fine-mapping to identify the putative causal variants and estimate the number of independent signals. We used Sum of Single Effect model (SuSiE)[24] to identify the credible set of putative causal variants, providing as input all variants with $P < 5 \times 10^{-8}$ from the meta-analysis results. For fine-mapping of signals identified in our meta-analysis of HIS and AFA, we used imputed genotype dosage for the same set of variants in MESA HIS and AFA, respectively. To select the parameter L (prior number of independent signals) for fine-mapping in SuSiE, DAP-G (Deterministic Approximation of Posteriors)[63] was conducted to provide a starting value for L based on the number of credible sets that the threshold of posterior inclusion probability was $>0.95$.

**Identification of novel versus previously reported signals.** To distinguish novel versus previously reported signals, we used the results from our previously published CHARGE GWAS n-3 ($n = 8866$)[19] and n-6 ($n = 8631$)[20] PUFAs in European ancestry to define the set of known signals. For each trait in the present GWAS effort, credible sets that included at least one variant reported in the previous CHARGE GWAS of the same trait in European ancestry were considered known, while the remaining signals were considered novel in the current study.

**Cross-ancestry replication analysis.** Following statistical fine-mapping, cross-ancestry replication analyses were conducted for the most highly supported putative causal variant from each credible set using data on n-3 and n-6 PUFAs from other race/ancestry groups. To do so, we examined results from the prior CHARGE GWAS meta-analysis of European American cohorts (CHARGE EUR), as well as GWAS results of HIS (CHARGE HIS) and AFA (CHARGE AFA) from the present study. As prior GWAS were performed using earlier imputation panels, we further used available measures of n-3 and n-6 PUFAs in self-reported European American (MESA EUR) and Chinese Americans (MESA CHN) from MESA as an additional source of replication having genotype imputation based on 1000 Genomes Phase 3, for consistency with our current work. The resources used for replication analyses were as follows. European Americans (MESA EUR and CHARGE EUR): 2344 self-reported European American participants from MESA (using 1000 Genomes Phase 3 imputation, for comparison with the current study), as well as summary statistics from the previously published CHARGE GWAS meta-analysis of n-3 ($n = 8866$)[19] and n-6 ($n = 8631$)[20] PUFAs based on imputation from the HapMap Phase I and II; African Americans (CHARGE AFA): summary statistics from the present GWAS of PUFAs in AFA to examine cross-ancestry replication of variants identified in the present GWAS of HIS; Hispanic Americans (CHARGE HIS): summary statistics from the present GWAS of PUFAs in HIS to examine cross-ancestry replication of variants identified in the present GWAS of AFA; and Chinese Americans (MESA CHN): 649 self-reported Chinese American participants from MESA (using 1000 Genomes Phase 3 imputation, for comparison with the current study).

The genetic association analyses performed for replication in each of these studies included covariate adjustment for age, sex, study site (where appropriate), and PCs of ancestry. Multiple testing correction was applied to account for the number of variants examined in cross-ancestry replication (HIS: $P < 0.05/19 = 0.0026$ and AFA: $P < 0.05/11 = 0.004$).

**Validation analysis.** Given the limited number of cohorts available for ancestry-specific and cross-ancestry replication of PUFA traits, additional validation analyses were conducted for the same set of variants using multi-ancestry genetic association with lipid traits (HDL, LDL, total cholesterol, and triglycerides) from the Global Lipids Genetics Consortium (GLGC)[25]. The GLGC aggregated GWAS results of lipid traits from 1,654,960 individuals from 201 primary studies. The genetic ancestry groups include admixed African or African, East Asian, European, Hispanic, and South Asian. The genetic analyses performed by GLGC included covariate adjustment for age, age$^2$, PCs of ancestry and any necessary study-specific

covariates. Multiple testing correction was applied to account for the number of variants examined in cross-ancestry validation (HIS: $P < 0.05/19 = 0.0026$ and AFA: $P < 0.05/11 = 0.004$).

**Bayesian colocalization analysis.** Bayesian colocalization analysis has proven an effective approach for the identification of downstream genes underlying GWAS loci[35]. We used the R/coloc package to conduct Bayesian colocalization analysis[64] to identify the putative gene(s) corresponding to each credible set of variants using MESA multi-ancestry eQTL data from purified monocytes[26] and GTEx multi-ancestry whole blood tissue eQTL data[65]. Bayesian colocalization analysis tested the following hypotheses: H0. neither GWAS of PUFAs nor eQTL has a genetic association in the region (within 1 Mb of the transcription start site); H1. only GWAS of PUFAs has a genetic association in the region; H2. only eQTL has a genetic association in the region; H3. both GWAS of PUFAs and eQTL are associated, but with different causal variants; H4. both GWAS of PUFAs and eQTL are associated and share a single causal variant. Colocalization for variants in credible sets was defined by (1) a posterior colocalization probability of hypothesis 4 (PP.H4) $> 0.80$, or (2) a PP.H4 $> 0.50$ and the ratio of PP.H4 / PP.H3 $> 5$.

**PrediXcan model.** PrediXcan, a gene-based association method focused on identifying trait-associated genes by quantifying the effect of gene expression on the phenotype on interest[66]. In this study, we applied summary-statistics based PrediXcan (S-PrediXcan)[30] using reference gene expression prediction models from MESA purified monocytes[26] and GTEx multi-ancestry whole blood[30]. S-PrediXcan associations were considered genome-wide significant if they passed the multiple testing correction for all genes (MESA: $P < 0.05/4470 = 0.00001$ and GTEx: $P < 0.05/4350 = 0.00001$).

**Chromatin contact analysis.** To identify variants located in open chromatin regions in contact gene promoters, we used GenomicRanges (v. 1.46.1; R version 4.1.1) to intersect the genomic coordinates (hg19) of the variants contained in the credible sets with the open chromatin peaks (called using the ENCODE pipeline) in significantly enriched contact with gene promoter determined by Promoter Capture C (Chicago Score $> 5$). We queried chromatin accessibility and promoter contacts in human mesenchymal stem cells (hMSC) and Adipocytes differentiated in vitro from these (hMSC_Adipocytes), embryonic stem cell derived hypothalamic neurons (hESC Hypothalamic Neurons), induced pluripotent-dervived Heptocytes (IPS-Hepatocytes), Enteroids, and the hepatic carcinoma HepG2 cell line[67–72].

**Gene co-expression analysis.** We used the GTEx whole blood gene expression version 8 TPM dataset to examine co-expression with *FADS1* for genes identified by integrative analyses, including colocalization and PrediXcan. Two models for gene co-expression analysis were used for each expression trait of interest: Model 1 - an unadjusted model *FADS1* ~ gene expression; and Model 2 - a covariate adjusted model *FADS1* ~ age + gender + gene expression.

Gene co-expression associations were considered statistically significant if they passed the multiple testing correction for all genes examined from colocalization and PrediXcan ($P < 0.05/39 = 0.0012$).

**Gene set enrichment analysis.** We applied gene set enrichment analysis for the combined set of genes identified by our integrative analyses (colocalization and PrediXcan) using the Molecular Signature Database (MSigDB) including hallmark gene sets (H), curated gene sets (C2), regulatory target gene sets (C3), computational gene sets (C4), ontology gene sets (C5), oncogenic signature gene sets (C6), immunologic signature gene (C7), cell type signature gene sets (C8)[31–33].

**Statistics and reproducibility.** Throughout the manuscript, statistical analyses and reported sample sizes reflect the number of biologically independent samples, with no single individual (person) contributing more than one data point to any given analysis. All P-values are presented based on two-sided statistical tests.

**Ethical review.** All relevant ethical regulations were followed for the study of human participants. All MESA, FHS and CHS participants provided written informed consent for participation at their respective study sites, including consent to participate in genetic studies. The MESA, FHS and CHS studies were also reviewed and approved by the Institutional Review Boards (IRBs) at each of the participating study sites. The current investigation including genetic analysis of n-3 and n-6 PUFA levels was reviewed and approved by the Institutional Review Boards (IRB) at the University of Virginia, the University of Washington and the Fatty Acid Research Institute.

**Reporting summary**. Further information on research design is available in the Nature Portfolio Reporting Summary linked to this article.

## Data availability

Genome-wide genotype data for the Multi-Ethnic Study of Atherosclerosis (MESA), the Framingham Heart Study (FHS) and the Cardiovascular Health Study (CHS) are available by application through dbGaP. The dbGaP accession numbers are: MESA phs000209, FHS phs000007, and CHS phs000287. Summary statistics resulting from our GWAS meta-analysis as presented in this manuscript are available on the CHARGE Summary Results site by application through dbGaP under the accession number phs000930. Summary statistics from the prior CHARGE GWAS of n-3 and n-6 fatty acids[19,20] were obtained from the CHARGE Consortium Results site[73]. Summary statistics from the GLGC GWAS of lipid levels[25] are available publicly[74]. Source data underlying Fig. 3b are presented in Supplementary Data 15. All other data are available from the corresponding author (or other sources, as applicable) on reasonable request.

## Code availability

Statistical fine-mapping of GWAS loci was conducted using SuSiE[24] as implemented using susieR version 0.12.27[75]. DAP-G[63] was used to choose the starting values for SuSiE and implemented using DAP-G version 1.0.0[76]. Bayesian colocalization analysis[64] was implemented using R/coloc version 5.1.0.1[77]. S-PrediXcan analysis was implemented using S-PrediXcan version 0.6.11[78]. Gene set enrichment analysis was implemented using MSigDB v7.5.1[79].

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

## Acknowledgements

This work was supported by R01 HL105756 to the CHARGE Consortium, NCCIH R01 AT008621 (to F.H.C.), and by the Daniel B. Burke Endowed Chair for Diabetes Research (to S.F.G.). The authors acknowledge Research Computing at The University of Virginia for providing computational resources and technical support that have contributed to the results reported within this publication. *The Multi-Ethnic Study of Atherosclerosis:* MESA and the MESA SHARe project are conducted and supported by the National Heart, Lung, and Blood Institute (NHLBI) in collaboration with MESA investigators. Support for MESA is provided by contracts HHSN268201500003I, N01-HC-95159, N01-HC-95160, N01-HC-95161, N01-HC-95162, N01-HC-95163, N01-HC-95164, N01-HC-95165, N01-HC-95166, N01-HC-95167, N01-HC-95168, N01-HC-95169, UL1-TR-000040, UL1-TR-001079, UL1-TR-001420, UL1-TR-001881, and DK063491. Funding for SHARe genotyping was provided by NHLBI Contract N02-HL-64278. Genotyping was performed at Affymetrix (Santa Clara, California, USA) and the Broad Institute of Harvard and MIT (Boston, Massachusetts, USA) using the Affymetrix Genome-Wide Human SNP Array 6.0. *The Cardiovascular Health Study:* This research was supported by contracts HHSN268201200036C, HHSN268200800007C, HHSN268201800001C, N01HC55222, N01HC85079, N01HC85080, N01HC85081, N01HC85082, N01HC85083, N01HC85086, 75N92021D00006, and grants U01HL080295 and U01HL130114 from the National Heart, Lung, and Blood Institute (NHLBI), with additional contribution from the National Institute of Neurological Disorders and Stroke (NINDS). Additional support was provided through R01AG023629 from the National Institute on Aging (NIA). A full list of principal CHS investigators and institutions can be found at CHS-NHLBI.org. Fatty acid measurements were supported by R01 HL085710. The provision of genotyping data was supported in part by the National Center for Advancing Translational Sciences, CTSI grant UL1TR000124, and the National Institute of Diabetes and Digestive and Kidney Disease Diabetes Research Center (DRC) grant DK063491 to the Southern California Diabetes Endocrinology Research Center. The content is solely the responsibility of the authors and does not necessarily represent the official views of the National Institutes of Health.. Additional support for infrastructure was provided by R01HL105756 and additional genotyping among the African-American cohort was supported in part by R01HL085251. DNA handling and genotyping at Cedars-Sinai Medical Center was supported in part by National Center for Research Resources grant UL1RR033176, now at the National Center for Advancing Translational Technologies CTSI grant UL1TR000124; in addition to the National Institute of Diabetes and Digestive and Kidney Diseases grant P30DK063491 to the Southern California Diabetes Endocrinology Research Center. *The Framingham Heart Study:* This research was partially supported by the National Human Genome Research Institute (R15HG006915; R01HL152215; NT). The Framingham Heart Study is conducted and supported by the National Heart, Lung, and Blood Institute (NHLBI) in collaboration with Boston University (Contract No. N01-HC-25195, HHSN268201500001I and 75N92019D00031). This manuscript was not prepared in collaboration with investigators of the Framingham Heart Study and does not necessarily reflect the opinions or views of the Framingham Heart Study, Boston University, or NHLBI. Data for this project was obtained via dbGaP (phs000007.v32.p1). Funding to support the Omni cohort recruitment, retention and examination was provided by NHLBI Contract N01-HC-25195, HHSN268201500001I and 75N92019D00031, as well as NHLBI grants R01-HL070100, R01-HL076784, R01-HL49869, and U01-HL-053941. Funding for SHARe Affymetrix genotyping was provided by NHLBI Contract N02-HL64278. SHARe Illumina genotyping was provided under an agreement between Illumina and Boston University. Funding for Affymetrix genotyping of the FHS Omni cohorts was provided by Intramural NHLBI funds from Andrew D. Johnson and Christopher J. O'Donnell. Funding support for the Framingham Red Blood Cell Fatty Acids dataset was provided by NHLBI grant R01 HL089590.

## Author contributions

Y.D.I.C., L.M.S., C.D.B., D.S., M.Y.T., S.S.R., D.M., and S.F.A.G. generated the data. C.Y., J.V., T.M.B., M.C.P., J.W., L.M.S., S.S.R., C.E.S., T.D.O., D.M., S.F.A.G., N.L.T., R.N.L., and A.M. analyzed the data. C.Y., L.M.S., S.S.R., C.E.S., D.M., N.L.T., R.N.L., and A.M. conceptualized and designed the study. C.D.B., M.Y.T., A.C.W., S.S.R., C.E.S., D.M., S.F.A.G., N.L.T., R.N.L., and A.M. provided critical oversight to data collection and study coordination. C.Y., M.C.P., B.H., S.S.R., T.D.O., F.H.C., N.L.T., R.N.L., and A.M. wrote the manuscript. All authors contributed to critical editing of the manuscript.

## Competing interests

The authors declare no competing interests.

## Additional information

[1]Center for Public Health Genomics, University of Virginia, Charlottesville, VA, USA. [2]Department of Biochemistry and Molecular Genetics, University of Virginia, Charlottesville, VA, USA. [3]Departments of Biology and Statistics, Dordt University, Sioux Center, IA, USA. [4]Department of Biostatistics, University of Washington, Seattle, WA, USA. [5]Cardiovascular Health Research Unit, Department of Medicine, University of Washington, Seattle, WA, USA. [6]Center for Spatial and Functional Genomics, The Children's Hospital of Philadelphia, Philadelphia, PA 19104, USA. [7]Division of Human Genetics, The Children's Hospital of Philadelphia, Philadelphia, PA, USA. [8]Center for Biomedical Informatics and Biostatistics, University of Arizona, Tucson, AZ, USA. [9]Institute for Translational Genomics and Population Sciences and Department of Pediatrics, The Lundquist Institute for Biomedical Innovation at Harbor-UCLA Medical Center, Torrance, CA, USA. [10]Fatty Acid Research Institute, Sioux Falls, SD, USA. [11]Division of Epidemiology and Community Health, University of Minnesota School of Public Health, Minneapolis, MN, USA. [12]Department of Genetics, Perelman School of Medicine, University of Pennsylvania, Philadelphia, PA, USA. [13]Institute for Biomedical Informatics, Perelman School of Medicine, University of Pennsylvania, Philadelphia, PA, USA. [14]New York Academy of Medicine, New York, NY, USA. [15]Department of Laboratory Medicine and Pathology, University of Minnesota, Minneapolis, MN, USA. [16]USDA/ARS Children's Nutrition Research Center, Baylor College of Medicine, Houston, TX, USA. [17]Nutrition and Genomics Laboratory, JM-USDA Human Nutrition Research Center on Aging at Tufts University, Boston, MA, USA. [18]Institute for Genome Sciences; Program in Personalized and Genomic Medicine; Department of Medicine, University of Maryland School of Medicine, Baltimore, MD, USA. [19]Friedman School of Nutrition Science & Policy, Tufts University, Tufts School of Medicine and Division of Cardiology, Tufts Medical Center, Boston, MA, USA. [20]Department of Pediatrics, Perelman School of Medicine, University of Pennsylvania, Philadelphia, PA, USA. [21]Division of Endocrinology and Diabetes, The Children's Hospital of Philadelphia, Philadelphia, PA, USA. [22]School of Nutritional Sciences and Wellness and the BIO5 Institute, University of Arizona, Tucson, AZ, USA. [23]University of Illinois, Chicago, Chicago, IL, USA. ✉email: amanicha@virginia.edu

