## [Peer Review File · Communications Biology]

Reviewers' comments:

Reviewer #1 (Remarks to the Author):

There are two major goals of the study. The first one is to examine whether the major loci identified in European Americans are shared across race/ancestry groups. The second one is to examine the evidence for genetic association unique to HIS and AFA populations.

Major comment:

1. The EAF between the discovery set and replication set is different a lot. For example in Table S6, the EAF of rs174585 is 0.061 for AA but 0.275 for Hispanic. How to explain these differences?
2. This study performed cross-ancestry replication analysis in CHARGE and MESA, with validation using the multi-ancestry GWAS of lipids from the Global Lipids Genetics Consortium (GLGC). In the method section, I cannot easily follow this part.
3. In the Follow-Up Replication and Validation Analyses of the method section, this study seems to use summary statistics from different cohort studies. How do you compare the SNP signals found in different sample sizes and adjusted variables?
4. In Replication and Validation Analyses, do you consider performing enrichment analysis?

Reviewer #2 (Remarks to the Author):

Review for "Genome-wide association studies and fine-mapping of genomic loci for n-3 and n-6 polyunsaturated fatty acids in Hispanic American and African American cohorts"

This is a very well-written manuscript investigating whether previously identified loci associated with n-3 and n-6 PUFAs in European cohorts are shared in other racial/ancestral groups, specifically African American and Hispanic American populations. A second purpose was to examine evidence for genetic associations that are unique to African American and Hispanic American populations. The authors confirmed the PUFA-associated signals that were previously identified in the European cohort (FADS1 and 2, PDXDC1, GCKR – both AFA and HIS; ELOVL2 AFA only). Additionally, novel signals were identified for arachidonic acid in the Hispanic American cohort (TMX2, SLC29A2, ANKRD13D, and POLD4). A POLD4 missense variant for arachidonic acid was identified and found to be unique to Hispanic American participants.

There is a great need for genetic association studies in non-European ancestry cohorts. This is a very well-written and interesting manuscript. The methods are well-described, and the results are clear and presented in a logical sequence. Tables and figures are also presented nicely. There are a few comments/suggestions regarding the figures and Discussion section that the authors may consider:

1. The text in the figures (Figures 2 and 3) is blurry and difficult to read, both when printed out and on the computer. If possible, would recommend trying to make the text in these figures clearer. They are an excellent part of the paper, and I would have liked to read them in detail.
2. A diagram clearly demonstrating the flow of the study and which cohorts/participants were used at each stage would be helpful for the reader to follow along. (Even if is just a supplemental figure).
3. The Discussion section of the paper is overall well-written. However, there are times that it seemed the results were simply being restated and summarized. A deeper interpretation of these results, and the implications they have would greatly benefit this manuscript. This is something that could simply be elaborated on at the end of the Discussion. For example, the authors state: "Our findings provide new insight into the complex genetics of circulating PUFA levels that reflect, in part, their response to evolutionary pressures across the course of human history." With respect to human health and disease, what are the implications of these findings? It would also be useful to elaborate on future work. The authors state clearly that there is a continued need to expand GWAS studies in cohorts with genetic ancestry that is reflective of the general population. But aside from this, and given the results

found in the present manuscript, what would be the next steps in this line of research? This is also something that could be elaborate on in the second to last paragraph of the Discussion section, or briefly mentioned in the last paragraph of the Discussion section.

4. The author's attempt at replication due to there being limited data for these traits is appreciated, but the replication scheme is difficult to understand.

5. Table 4 – please clarify which race group the discovery falls under and what race the replication results come from.

Reviewers' comments:

Reviewer #1 (Remarks to the Author):

There are two major goals of the study. The first one is to examine whether the major loci identified in European Americans are shared across race/ancestry groups. The second one is to examine the evidence for genetic association unique to HIS and AFA populations.

Response: We thank the reviewer for the targeted synthesis of our manuscript and identification of points for improvement.

Major comment:

1. The EAF between the discovery set and replication set is different a lot. For example in Table S6, the EAF of rs174585 is 0.061 for AA but 0.275 for Hispanic. How to explain these differences?

Response: We agree that the EAF of some variants differ between the discovery and replication cohorts. The reason of the EAF differences is that we applied cross-ancestry replication analysis and some of the variants examined for replication show distinct allele frequencies across ancestry groups. We highlight the point about cross-ancestry differences in allele frequencies in the fourth paragraph of the section on replication/validation in the Results: "Some of the novel signals without cross-ancestry replication demonstrated large differences in allele frequencies across groups..." (p. 12).

2. This study performed cross-ancestry replication analysis in CHARGE and MESA, with validation using the multi-ancestry GWAS of lipids from the Global Lipids Genetics Consortium (GLGC). In the method section, I cannot easily follow this part.

Response: We thank the reviewer for this valuable comment. To improve clarity, we have now separated the 'Follow-up replication and validation analyses' paragraph into two paragraphs ('Cross-ancestry replication analysis' and 'Validation analysis') and added additional information to clarify the replication analysis and validation analysis (pp. 22-23).

3. In the Follow-Up Replication and Validation Analyses of the method section, this study seems to use summary statistics from different cohort studies. How do you compare the SNP signals found in different sample sizes and adjusted variables?

Response: We thank the reviewer for this insightful comment. We agree with the reviewer that it is important to check the consistency of covariate adjustments used in the discovery and replication/validation analyses.

The cross-ancestry replication analyses examining associations with fatty acid traits were performed by the CHARGE consortium investigators, including ourselves, and used covariate adjustments completely consistent with those presented in our discovery analyses. Specifically, these analyses were all adjusted for age, sex, study site (where appropriate) and PCs of ancestry.

For the validation analysis based on the lipid traits from GLGC resources, analyses were carried out separately by ancestry group at each cohort and residuals were generated separately in males and females adjusting for age, age², PCs of ancestry and any necessary study-specific covariates (Graham *et al.*, PMID: 34887591). Thus, the covariate adjustments used in the GLGC paper are consistent with those incorporated in our discovery analysis of fatty acid traits

We have now added to the manuscript information regarding covariate adjustments for each of the replication and validation analyses (p. 23).

Regarding the sample sizes used for replication and validation, we agree with the reviewer that these analyses exhibit differences in sample size compared to our discovery analyses. Indeed, we chose the GLGC for our validation effort as it is the largest GWAS of lipid levels to date, and we expected it to be well-powered to demonstrate associations with the selected phenotypes. We now include text explaining the motivation of our selection of replication and validation cohorts in the Discussion (pp. 15-16).

4. In Replication and Validation Analyses, do you consider performing enrichment analysis?

Response: We thank the reviewer for this thoughtful suggestion. We now include the results of enrichment analysis for the list of genes identified by our colocalization and PrediXcan analyses (see Table S14). Also, we added description of our approach for the enrichment analysis in the Methods section (p. 25), brief summary of those findings in the Results section (p. 14), and further interpretation of those results in the Discussion (p. 18).

Reviewer #2 (Remarks to the Author):

Review for “Genome-wide association studies and fine-mapping of genomic loci for n-3 and n-6 polyunsaturated fatty acids in Hispanic American and African American cohorts”

This is a very well-written manuscript investigating whether previously identified loci associated with n-3 and n-6 PUFAs in European cohorts are shared in other racial/ancestral groups, specifically African American and Hispanic American populations. A second purpose was to examine evidence for genetic associations that are unique to African American and Hispanic American populations. The authors confirmed the PUFA-associated signals that were previously identified in the European cohort (FADS1 and 2, PDXDC1, GCKR – both AFA and HIS; ELOVL2 AFA only). Additionally, novel signals were identified for arachidonic acid in the Hispanic American cohort (TMX2, SLC29A2, ANKRD13D, and POLD4). A POLD4 missense variant for arachidonic acid was identified and found to be unique to Hispanic American participants.

There is a great need for genetic association studies in non-European ancestry cohorts. This is a very well-written and interesting manuscript. The methods are well-described, and the results are clear and presented in a logical sequence. Tables and figures are also presented nicely.

Response: We thank the reviewer for the careful reading of our manuscript and appreciation of our motivation and key points.

There are a few comments/suggestions regarding the figures and Discussion section that the authors may consider:

1. The text in the figures (Figures 2 and 3) is blurry and difficult to read, both when printed out and on the computer. If possible, would recommend trying to make the text in these figures clearer. They are an excellent part of the paper, and I would have liked to read them in detail.

Response: We thank the reviewer for recognizing the value of our figures and the need for improvement. We have re-generated the figures to improve resolution and improve readability of the axis labels (now Figures 3 and 4).

2. A diagram clearly demonstrating the flow of the study and which

cohorts/participants were used at each stage would be helpful for the reader to follow along. (Even if is just a supplemental figure).

Response: We thank the reviewer for this helpful suggestion. We have now added a diagram to demonstrate the overall approach of our study (Figure 2).

3. The Discussion section of the paper is overall well-written. However, there are times that it seemed the results were simply being restated and summarized. A deeper interpretation of these results, and the implications they have would greatly benefit this manuscript. This is something that could simply be elaborated on at the end of the Discussion. For example, the authors state: “Our findings provide new insight into the complex genetics of circulating PUFA levels that reflect, in part, their response to evolutionary pressures across the course of human history.” With respect to human health and disease, what are the implications of these findings? It would also be useful to elaborate on future work. The authors state clearly that there is a continued need to expand GWAS studies in cohorts with genetic ancestry that is reflective of the general population. But aside from this, and given the results found in the present manuscript, what would be the next steps in this line of research? This is also something that could be elaborate on in the second to last paragraph of the Discussion section, or briefly mentioned in the last paragraph of the Discussion section.

Response: We thank the reviewer for careful reading of our Discussion and constructive comments for improvement. We have added to the end of the Discussion to emphasize the broader implications of our work in terms of the biological pathways identified (p. 18), as well as precision nutrition and other biological implications (p. 19).

4. The author’s attempt at replication due to there being limited data for these traits is appreciated, but the replication scheme is difficult to understand.

Response: We thank the reviewer for this comment which is also in agreement with the feedback we received from Reviewer #1. We have now re-organized and expanded the Methods text to improve overall clarity the sections focused on replication and validation analysis (pp. 22-23).

5. Table 4 – please clarify which race group the discovery falls under and what race the replication results come from.

Response: We thank the reviewer for identifying this point for clarification. We have now improved Table 4 by adding a new column with detailed information from the discovery analysis, as well as race/ancestry information for the replication results.

REVIEWERS' COMMENTS:

Reviewer #1 (Remarks to the Author):

This is a very well-written manuscript. I have no more questions.

Reviewer #2 (Remarks to the Author):

A well-written manuscript. All concerns from the first review have been addressed. No further comments at this time.